# Evaluating Material’s Interaction in Wire Electrical Discharge Machining of Stainless Steel (304) for Simultaneous Optimization of Conflicting Responses

**DOI:** 10.3390/ma12121940

**Published:** 2019-06-17

**Authors:** Kashif Ishfaq, Naveed Ahmad, Muhammad Jawad, Muhammad Asad Ali, Abdulrahman M. Al-Ahmari

**Affiliations:** 1Department of Industrial and Manufacturing Engineering, University of Engineering and Technology, Lahore 54890, Pakistan; naveed527@gmail.com; 2Department of Industrial Engineering, University of Engineering and Technology, Taxila 47080, Pakistan; engr.jawad@uettaxila.edu.pk (M.J.); asadaliuetian@gmail.com (M.A.A.); 3Raytheon Chair for Systems Engineering, Advanced Manufacturing Institute, King Saud University, Riyadh 11421, Saudi Arabia; alahmari@ksu.edu.sa

**Keywords:** stainless steel (304), grey relational analysis, surface quality, ANOVA, cutting rate, kerf

## Abstract

Stainless steel (SS 304) is commonly employed in industrial applications due to its considerable corrosion resistance, thermal resistance, and ductility. Most of its intended applications require the formation of complex profiles, which justify the use of wire electrical discharge machining (WEDM). However, its high thermal resistance imposes a limitation on acquiring adequate surface topography because of the high surface tension of the melt pool, which leads to the formation of spherical modules; ultimately, this compromises the surface quality. Furthermore, the stochastic nature of the process makes it difficult to optimize its performance, especially if more than one conflicting response is involved, such as high cutting speed with low surface roughness and kerf width. Therefore, this study aimed to comprehensively investigate the interaction of SS 304 and WEDM, with a prior focus on simultaneously optimizing all the conflicting responses using the Taguchi-based grey relational approach. Analysis of variance (ANOVA) revealed that the current was the most significant parameter for cutting speed and kerf, whereas roughness, voltage (45%), drum speed (25.8%), and nozzle offset distance (~21%) were major contributing factors. SEM micrographs showed that optimal settings not only ensured simultaneous optimization of the conflicting responses but also reduced the number and size of spherical modules.

## 1. Introduction

Wire electrical discharge machining (WEDM) is a nonconventional machining process in which a cutting action is performed by electrical sparking between a work piece and an electrode. The spark temperature ranges from 8000 to 12,000 °C. Such a high temperature melts and vaporizes the workpiece material [1]. The amount of material erosion is governed by the thermal conductivity of the material [2]. The gap between the workpiece and the electrode is usually maintained between 0.025 and 0.075 mm [3,4]. WEDM is commonly used to cut hard and high-strength materials such as tool steel, stainless steel, titanium, and metal composites with precise dimensions and a good surface finish. Therefore, there is no need for postmachining operations such as grinding and polishing. WEDM has many applications in the tool- and die-making and prototype-manufacturing industries [5,6,7,8,9,10]. 

Stainless steel (SS 304) is used in numerous applications such as cryogenic vessels, evaporators, dies, molds, valves, and refrigerant equipment because of its high ductility, good corrosion and thermal resistance, and nonmagnetic nature [11,12]. Most of the intended applications of this material require the formation of complex profiles which necessitate the use of WEDM.

In WEDM, the quality of the parts, dimensional accuracy, and process economics are determined by surface roughness (SR), kerf width (KW), and cutting speed (CS), respectively. These response characteristics are greatly affected by several input parameters, including pulse on-time (Ton), pulse off-time (Toff), current (I), servo voltage (V), and so forth [13,14]. Mandal et al. [15] optimized the cutting speed and surface roughness of the Nimonic C-263 super alloy using WEDM. It was concluded that Ton, Toff, and voltage significantly affected the cutting speed and wire wear rate, while Toff had little impact on SR. Yang et al. [16] investigated the effect of WEDM parameters on the machining of tungsten. It was found that by increasing the Ton value, the corresponding surface roughness decreased, whereas cutting speed was observed to be greatly affected by servo voltage, wire tension, and wire feed rate. Azam et al. [17] reported that pulse off-time, power, and pulse frequency are the major parameters that affect cutting speed. Optimized cutting speed was observed at a low pulse off-time value, high power, and high pulse frequency when machining high speed low alloy steel (HSLA steel). Ikram et al. [18] investigated the effect of different variables on the SR, KW, and material removal rate (MRR) of tool steel (D2 type). Pulse on-time, voltage, and wire tension were found to be significant control parameters. A better surface was obtained at low levels of pulse on-time and voltage. Minimum kerf width was achieved at a low level of pulse on-time and a high level of voltage. Huang et al. [1] observed the influence of different machining conditions on the WEDM of SKD11 alloy steel and concluded that bed speed and pulse on-time had a noticeable influence on MRR, while pulse on-time was the only significant parameter for SR and gap width. Shah et al. [19] investigated tungsten carbide samples machined via WEDM considering seven input parameters along with material thickness. It was reported that wire tension and pulse on-time had more influence on kerf width compared with other factors. It was also noticed that an increase in wire tension resulted in a reduction of kerf width. Kumar et al. [20] worked on the optimization of WEDM parameters for high-speed steel. It was found that peak current and pulse on-time were inversely proportional to the surface finish. Peak current, pulse on- and off-time, and wire feed were found to be the significant control variables, while flushing pressure was observed as insignificant for SR. Dhobe et al. [21] analyzed the effects of four WEDM parameters on SR when machining cryo-treated AISI D2 tool steel. It was observed that pulse on-time, peak current, and voltage had a significant impact on SR, while the impact of pulse off-time was insignificant for this response. Bhatia et al. [22] evaluated the effect of WEDM input parameters on surface roughness while machining high-carbon Cr steel using the WEDM process. Brass wire was used in experiments. The experimental results revealed that surface roughness was significantly affected by changing pulse off-time. In another study the potential of WEDM was tested for cutting [23] T90 Mn2 Cr45W50 tool steel considering Toff, Ton, peak current, and wire tension as the control variables. It was concluded that peak current was directly proportional to the surface roughness of the machined specimen. Bobbili et al. [24] studied the effects of WEDM parameters on SR of high-strength armor steel, they reported that Toff, Ton, and spark voltage were the significant variables for the selected response. Zhang et al. [25] studied the effects of input parameters on the WEDM of SKD11 alloy tool steel. The experimental data revealed that Ton and Toff were the most significant parameters for SR. A poor surface was achieved at high values of Ton and wire speed. Another work found that [26] the cutting width was increased by increasing the value of Ton. It was also observed that the kerf width at the top surface was larger than the kerf width at the bottom side. It was also noticed that the difference was increased by raising the magnitude of Ton. Furthermore, it was also found that higher wire tension reduced the wire efficiency, which ultimately resulted in low SR. Tilekar et al. [27] conducted an experiment on aluminum and mild steel using WEDM. The results indicated that wire feed and Ton had a significant impact on kerf width for aluminum and mild steel, respectively.

Researchers have used several WEDM input parameters, such as pulse on-time, pulse off-time, wire tension, wire feed rate, current, and servo voltage, on different materials to optimize the various output responses. However, the effect of nozzle offset distance (N_OD_) and drum speed (DS) during WEDM of SS 304 has not been specifically studied, which is the research topic of the present work. It is also worth noting that previous studies have mainly focused on single-response optimization, which may not serve the purpose if there are more response characteristics involved, especially conflicting ones such as high cutting speed with minimum surface roughness and kerf width. Machinists have been searching for an optimal solution that simultaneously optimizes all the response attributes—most importantly, the abovementioned responses. Therefore, in this study, the optimal combination of parameters was developed using the grey relational approach (GRA), which simultaneously optimized the considered responses (cutting rate, surface roughness, and kerf width).

## 2. Materials and Methods 

WEDM (DK7735) was used for the experiments, which were conducted on an SS 304 workpiece with a density of 8.04 mg/m^3^ and a Rockwell hardness of 68 HRB. The experimental setup included molybdenum wire (diameter of 0.18 mm and tensile strength of 883 GPa), a machine control unit, a power supply, and a computerized numeric control (CNC) generating system (HF), as shown in Figure 1. The material composition of SS 304 (Table 1) was obtained by spectroscopy using the ASTM E 167-4 standard procedure. 

Wire breakage is a common problem associated with WEDM. This issue not only hampers the cut quality but also prolongs machining time, which in turn adds to the cost. Therefore, during preliminary trials, this issue was considered and factor levels were chosen which minimized the chance of wire breakage. Four WEDM parameters—V, DS, I, and N_OD_—were selected to evaluate their impact on the output responses SR, CS, and KW. Surface roughness of the specimen was measured by using a surface texture meter (*Surtronic S128*, Taylor Hobson, Leicester, UK), manufactured by Taylor Hobbson, at an evaluation length of 4 mm and cutoff length of 0.8 mm. Three profiles on either side of the specimen were measured and then the average value of the surface roughness was reported. Kerf width was measured by a coordinate measuring machine (CMM: CE-450DV, CHIEN WEI Precise Technology Co., LTD. Taiwan) with a resolution of up to 0.001 mm. Cutting speed was measured by dividing the total distance traveled by the time consumed cutting the specimen. A Taguchi orthogonal L16 array of experiments was used to observe the effects of input parameters (voltage, drum speed, current, and nozzle offset distance) on the response parameters. Taguchi is a cost-effective statistical technique which reduces the number of experiments required to find the individual optimal output response with the optimum combination of input parameters [27,28]. 

Four input parameters with their respective levels are shown in Table 2. The selection of these parameters and their respective levels was based on the minimum chance of wire rupture. Moreover, those trials in which the wire was broken were not considered true, and thus, those runs were repeated. A total of 16 experiments were conducted according to the L16 orthogonal array. The conducted experiments were performed in a randomized order. Each experimental trial was performed three times in order to ensure the reliability of the results. The standard deviation for all trials was also calculated and it was found that the results were closely spaced; therefore, the average values are reported herein. During experimentation, perpendicularity of the work surface with respect to the machine table was ensured with the aid of a set square. Workpiece dimensions after machining were 8 × 8 × 10 mm. Experimental results were analyzed using different statistical techniques, such as a parametric effect plot and ANOVA. All analyses were performed on Minitab 16 software. The optimal parametric combination was also developed using the GRA with the relations mentioned in Equations (1)–(5).

### 2.1. Grey Relational Generating

In this step, the original data sequence was normalized between 0 and 1 by using three types of data normalizing relations mentioned below—“nominal the better”, “lower the better”, and “higher the better”—depending upon the desired output of response.

Higher the better:(1)$$Xi\left(k\right)\;=\;\frac{Xi\left(k\right)-minXi\left(k\right)}{maxXi\left(k\right)-\;minXi}$$

Lower the better:(2)$$Xi*\left(k\right)\;=\;\frac{maxXi\left(k\right)-Xi\left(k\right)}{maxXi\left(k\right)-\;minXi\left(k\right)}$$

Nominal the better:(3)$$Xi*\left(k\right)\;=\;1-\frac{\left|Xi\left(k\right)-Xob\left(k\right)\right|}{maxXi\left(k\right)-\;Xiob\left(k\right)}$$
where *Xi*
∗
*(k)* is the normalized value of the *k*th element in the *i*th sequence, *X*_o*b*_*(k)* is the desired value of the *k*th quality characteristic, min *Xi*
∗
*(k)* is the smallest value of *Xi(k)*, max *Xi*
∗
*(k)* is the largest value of *Xi(k), n* is the number of runs, and *p* is the number of quality features. For the cutting speed, the larger the better relation was used, and for both the kerf width and surface roughness, the smaller the better relationship was employed. 

### 2.2. Grey Relational Coefficient (GC)

The second step was to calculate the GC, which was used to find the relationship between the optimal and the actual normalized output results. Equation (4) was used for this calculation:(4)$$\gamma \left(k\right)\;=\;\gamma \left(Xo\left(k\right)-Xi\left(k\right)\right)=\frac{\Delta min+\zeta \Delta max}{\Delta o,i\left(k\right)+\zeta \Delta max}$$
where ∆*min* and ∆*max* are the minimum and maximum absolute differences, respectively, which represent a deviation from the target value and can be treated as quality loss. *ζ* is the identification coefficient and its value ranges between 0 and 1. In this study, the value of this coefficient was assumed to be 0.5. 

### 2.3. Grey Relational Grade

The grey relational grade is the weighting sum of the GC. It was calculated using Equation (5):(5)$$\gamma \left(xo-xi\right)\;=\;\overset{n}{\underset{k=1}{\sum }}\beta k(xo-xi)$$
where *βk* represents the weighting value of the *k*th output response. A high grey relational grade value indicates a stronger relationship between the ideal sequence and the present sequence. The ideal sequence is the best response in the machining process. A higher grey grade indicates that the current sequence is closer to the desired sequence.

## 3. Results

This section briefly explains the statistical significance of the factors, which was identified by ANOVA, followed by parametric effect analysis, and finally, multiobjective optimization through GRA.

### 3.1. Analysis of Variance for SR, CS, and KW

ANOVA was used at a 95% confidence interval to check the significance of the input parameters with respect to the selected response attributes. The results of the analysis are shown in Table 3. Furthermore, the percentage contribution that depicts the influence of an input parameter on the response was also determined by ANOVA [29]. Based on ANOVA, it was found that the voltage, drum speed, and nozzle offset distance were significant parameters for surface roughness because their p-values were less than the predefined alpha value of 0.05. However, voltage, with a percentage contribution of 45%, was the leading contributing factor compared with other parameters, as illustrated in Figure 2a. Drum speed was the second most contributing parameter for surface roughness, followed by nozzle offset distance and discharge current, with percentage contributions of 20% and 7%, respectively. The results revealed that in the case of cutting speed, the current was the only significant input parameter, while all other parameters were insignificant for this response. The percentage contribution of the current with respect to cutting speed was exceptionally high (85%), as shown in Figure 2b. In the case of kerf width, the discharge current, voltage, and drum speed were observed to be the significant control factors as per the ANOVA results shown in Table 3. However, the current was the leading factor, with a percentage contribution of 53% (Figure 2c). Among the remaining factors, voltage was the second major contributing factor, with a percentage contribution of 22%, followed by drum speed, which had a percentage contribution of 21%.

### 3.2. Analysis of the Effect of Control Factors on Responses

After finding the significant parameters for the selected response attributes using ANOVA, parametric effect analysis was performed to determine the effect of the control variables on the set responses. The parametric effect plot shown herein also includes error bars of standard error to better understand the results. While analyzing the effects, only those factors are discussed which were rated significant as per ANOVA results for the selected output variables. 

#### 3.2.1. Effect of Current

As mentioned in the previous section, the discharge current was the only factor which was screened out as significant for cutting speed during WEDM of SS 304, with the highest percentage contribution of 85%. The main effects plot presented in Figure 3a shows that this factor had a direct relationship with the cutting speed (i.e., higher current resulted in higher cutting rate). In WEDM, discharge energy is primarily responsible for the material removal from the workpiece surface, and at higher value of current, a greater amount of discharge energy is available for cutting, which results in a faster cutting speed. Consequently, a faster cutting rate is achieved. Here, an increase in current from 1 to 4 A caused the cutting speed to rise from 0.99 to 2.24 mm/min (an approximately 126% increase). A similar trend for current with respect to cutting speed has also been reported for WEDM of HSLA steel [16]. Further, current also proved to be a significant control parameter for kerf width. Additionally, it had the highest percentage contribution (53%) for kerf width among the other input factors. The trend of this factor for kerf width was somewhat like that for cutting speed, which can be seen in Figure 3b. A higher current value tended to widen the machined kerf because a larger current led to higher explosions of energy, which resulted in intense sparking. This eventually produced deeper and wider craters. In this way, the kerf width increased as highlighted in Figure 4. A similar relationship between current and kerf width was also reported in [30]. In the case of surface roughness, the effect of current was statistically insignificant according to the ANOVA results, shown in Table 3, and is therefore not discussed here. 

#### 3.2.2. Effect of Voltage

Voltage was a significant factor for both surface roughness and kerf width, but its percentage contribution was higher in the case of surface roughness compared with kerf width. Voltage was a major contributing factor (45%) among the selected parameters for surface roughness (Figure 2a,c). As the voltage value increased, the corresponding kerf width reduced, as shown in Figure 5a. This reduction in kerf width is attributed to the fact that the wire electrode stayed a bit farther from the workpiece at higher voltage settings. Hence, less energy was transferred to the surface, as the workpiece–electrode gap was large. Therefore, a small amount of material was removed from the work specimen, which produced a narrower slot (i.e., smaller kerf width). There was about a 17% decrease in the kerf width as the voltage values increased from 50 to 80 V. The trend of this input parameter (voltage) for surface roughness was similar to that of kerf width (i.e., an increase in the voltage value reduced the surface roughness), as depicted in Figure 5b. The SR value reduced from 5.78 to 4.62 µm as the voltage changed from 50 to 80 V (an approximately 25% reduction in surface roughness). Ikram et al. [17] also stated that a good surface finish could be achieved at 80–85 V. Primarily, this reduction in SR was due to a decrease in the amount of discharge energy transferred to the work surface because a higher voltage keeps the wire electrode at a larger offset from the target surface. Subsequently, the workpiece bears less heat input, which results in the formation of shallow craters, which in turn improves the surface finish, as shown in the SEM images in Figure 6. The roughness profiles recorded for the two scenarios—one in which the voltage was set at the maximum value and the other in which the voltage was set at the lowest level—clearly show that surface finish was better in the second condition, as displayed in Figure 7. 

#### 3.2.3. Effect of Drum Speed 

DS was a significant parameter for surface roughness and kerf width. However, its role was more prominent in the case of surface roughness, as its percentage contribution was higher for this response (~26%). An increase in the DS value produced a rougher surface, as shown in Figure 8a. This reduction in surface finish is attributed to the contact duration between the wire and the workpiece surface, which was reduced at a higher drum speed. Basically, for better cut quality, the target surface should have uniform sparking for an adequate amount of time so that localized heating can effectively melt and vaporize the work surface. However, the increase in drum speed not only reduces the contact time but also has a detrimental effect on uniform sparking. As a result, a poor surface finish is obtained. The behavior of drum speed was noticed here to be reversed for kerf width (i.e., a larger kerf width was obtained at a lower drum speed, as shown in Figure 8b). The width of the kerf in WEDM was not only generated by localized heating from the exposed surface of the wire electrode to the target surface, but the wire was continuously vibrating throughout its machining action. The surface of the machined specimen was also affected by these induced vibrations [31]. These vibrations promoted the chance of sideways sparking, which also contributed to determining the kerf width. At a higher wire speed, the amplitude of sideways sparking reduced and, subsequently, a narrower kerf was produced. About a 10% reduction in kerf width was observed as the drum speed value rose from 35 to 50 Hz.

#### 3.2.4. Effect of Nozzle Offset Distance

The nozzle offset distance was observed to be the only significant control variable for surface roughness as per the ANOVA results (Table 3). It was the third most contributing factor for surface roughness, with a percentage contribution of about 21%. However, for the other two responses, its role was found to be insignificant. The trend of this parameter for surface roughness was similar to that of voltage for this response (i.e., increase in N_OD_ yielded a better surface finish, as shown in Figure 9). The surface roughness value decreased from 5.98 to 5.07 µm as the value of N_OD_ increased from 200 to 260 mm (an approximately 18% reduction in surface roughness). 

In WEDM, the role of this nozzle is to provide pressurized flushing medium in the machining area to remove melt debris. By changing the nozzle offset (distance between the flushing nozzle and the workpiece), the amount of flushing pressure borne by the target surface varies. At lower values of N_OD_, more flushing pressure is placed onto the melt pool of the machining regime, which causes a rapid quenching of the melt pool. The melt debris once again adheres to the base metal instead of being flushed away. The weldment pattern of the melt debris is not uniform, as this phenomenon happens abruptly. It yields a more irregular surface texture, due to which the surface roughness is increased. At a higher offset value, however, less flushing pressure is targeted on the work surface, which not only facilitates the uniform removal of material but also improves the surface finish. The SEM micrograph of machined sample at a lower N_OD_ value clearly shows that the machined surface was subjected to a large number of spherical modules, which indicates that the melt debris was not effectively flushed away (see Figure 10). Moreover, it was observed that the diameter of the spherical modules at the machined surface was also of larger magnitude. It has already been established that the SR of a machined specimen increases with an increase in the diameter of spherical modules [32]. Therefore, the surface finish reduces at a lower N_OD_ value. 

### 3.3. Multiresponse Optimization through GRA

GRA is a decision-making technique used to remove deficiencies in experimental results analyzed through statistical methods. It is an effective tool which overcomes the drawbacks of simple statistical methods with fewer data points [23,33]. The steps involved in GRA have already been described in the Materials and Methods section. For this work, the data were first normalized (grey relational generating) using three types of relations, as mentioned in Equations (1)–(3) (nominal the better, lower the better, and higher the better) depending upon the desired response output [34,35]. Grey relation generating values for all the experiments are shown in the Table 4. Afterwards, the GC, which was used to find the relationship between the optimal and the actual normalized output results, was calculated using Equation (4). In this study, the value of coefficient *ζ* was assumed to be 0.5, as reported by Azhiri et al. [36] and Julong [37]. As a third step, grey relational grades were found employing the relationship described in Equation (5). The GRA grade and ranking calculations are shown in Table 4. 

Based on the results, it was found that the GRA grade for the 14th trial was 0.7833, which was the highest. Thus, this experimental setting (voltage of 50 V, drum speed of 35 Hz, current of 3 A, and nozzle offset distance of 220 mm) represented the optimal combination of parameters that can simultaneously optimize all the selected response attributes. 

#### Confirmatory Test

In order to validate the proposed optimal setting of control parameters, a confirmatory experiment was performed. The results of the confirmatory test are provided in Table 5. It was found that the desired outcome of all the selected responses was optimized, as shown in Figure 11.

## 4. Conclusions

In this research, the effects of two commonly used parameters (current and voltage) and two rarely used parameters (drum speed and nozzle offset distance) were evaluated on surface roughness, kerf width, and cutting speed during WEDM of SS 304. Electrode–workpiece interaction was studied with the help of SEM-based evidence in terms of melt pool formation and debris exclusion from the electrode–workpiece gap. After employing traditional statistical tests to filter out the significant levels of parametric effects, grey relational analysis was also performed for multi-objective optimization. Based on the results and their analyses, the following conclusions may be drawn:Analysis of variance revealed that the voltage, drum speed, and nozzle offset distance were significant factors for surface roughness. However, voltage was the major contributing factor, with a percentage contribution of 45%, followed by drum speed (25.8%) and nozzle offset distance (~21%). Higher values of N_OD_ and V at a low DS yielded a high surface finish.Cutting speed during WEDM of SS 304 was mainly influenced by the current, which had an exceptionally high percentage contribution of 85.5%. Moreover, an increase in the current value had a positive impact on the cutting rate. The role of drum speed and nozzle offset distance was observed to be insignificant for cutting rate; however, smaller N_OD_ and larger DS values improved the cutting rate.In addition to current and voltage, drum speed was also found to be a contributing factor in reference to kerf width. The percentage contributions of current, voltage, and drum speed were 53.3%, 22.2%, and 21.2%, respectively, for kerf width. However, high values of drum speed and voltage along with a low amount of current yielded a narrower kerf.SEM analysis revealed that the cut surface was crowded with spherical modules at a larger nozzle offset distance because the flushing capability of the dielectric had been reduced. A low offset distance value ensured appropriate flushing because of the higher dielectric pressure, which ultimately minimized spherical module formation on the machined surface.Against conflicting response attributes such as surface roughness, cutting speed, and kerf width, the optimal combination of WEDM parameters achieved though grey relational analysis was voltage of 50 V, drum speed of 35 Hz, current of 3 A, and nozzle offset distance of 220 mm. This combination provided the maximum cutting speed (2.62 mm/min) along with the minimum amount of surface roughness (4.47 µm) and kerf width (0.32 mm). The results were validated by a confirmatory test, as presented in Table 5 and Figure 11.

## Figures and Tables

**Figure 1 materials-12-01940-f001:**
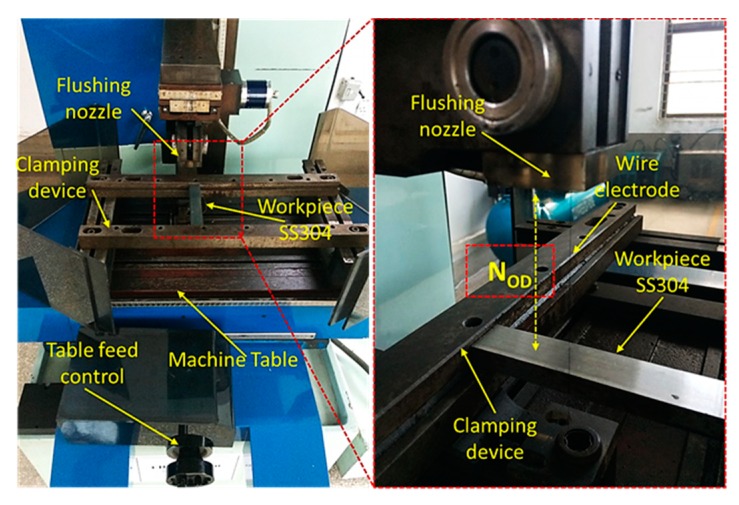
Experimental setup.

**Figure 2 materials-12-01940-f002:**
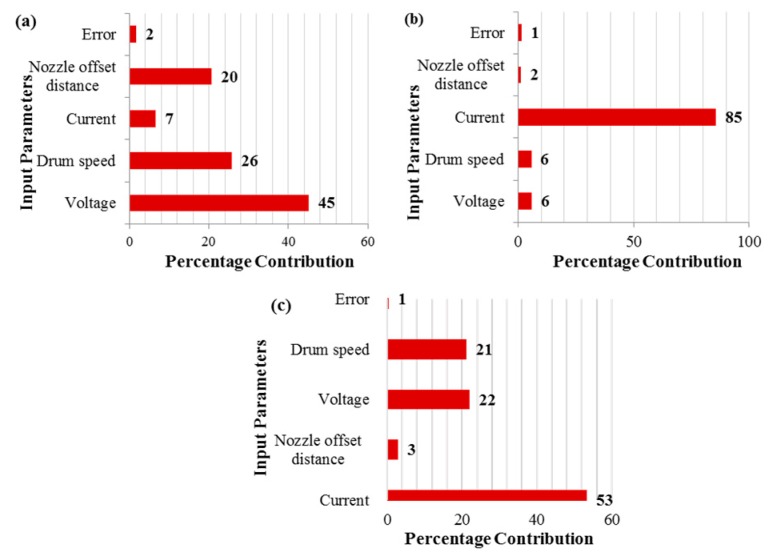
Percentage contribution of factors for (**a**) SR, (**b**) CS, and (**c**) KW.

**Figure 3 materials-12-01940-f003:**
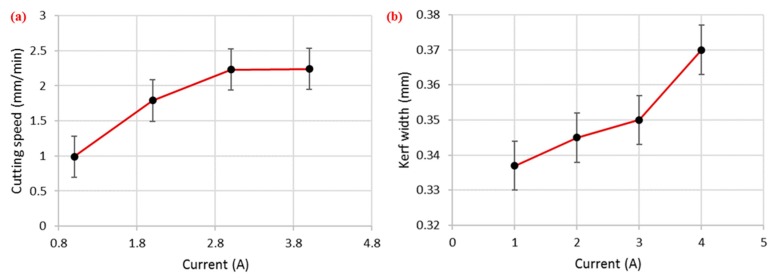
Effect of current on (**a**) cutting speed and (**b**) kerf width.

**Figure 4 materials-12-01940-f004:**
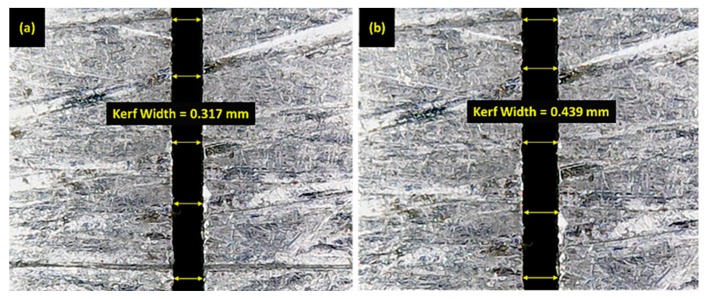
Kerf width at (**a**) 1 and (**b**) 4 A.

**Figure 5 materials-12-01940-f005:**
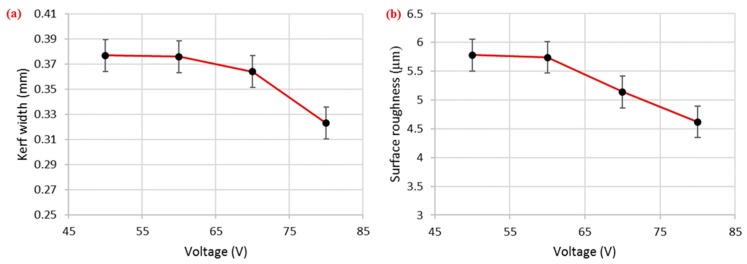
Effect of voltage on (**a**) kerf width and (**b**) surface roughness.

**Figure 6 materials-12-01940-f006:**
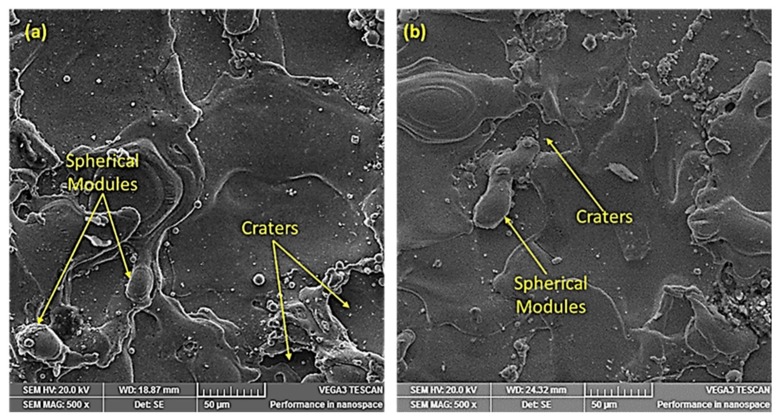
SEM micrograph of machined samples at: (**a**) 4 A, 50 V, 45 Hz, and 240 mm; and (**b**) 1 A, 80 V, 45 Hz, and 220 mm.

**Figure 7 materials-12-01940-f007:**
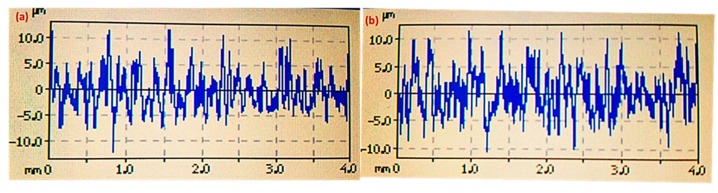
Surface roughness profiles of the machined surface at (**a**) 50 and (**b**) 80 V.

**Figure 8 materials-12-01940-f008:**
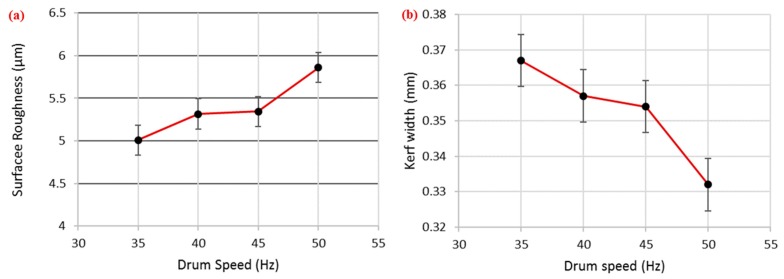
Effect of drum speed on (**a**) surface roughness and (**b**) kerf width.

**Figure 9 materials-12-01940-f009:**
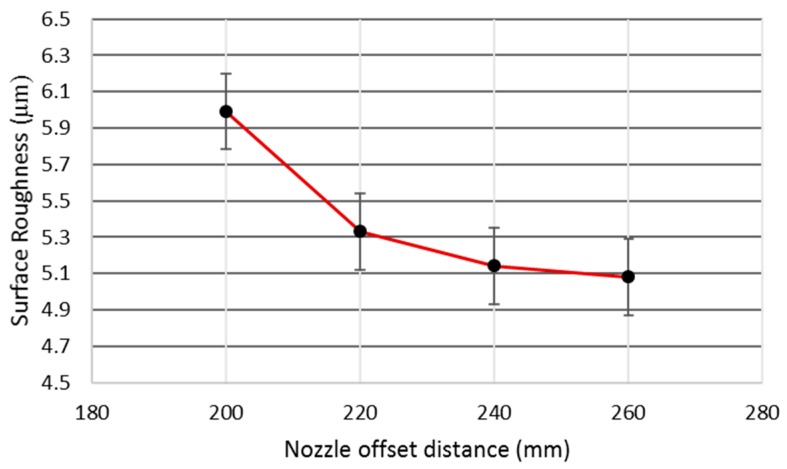
Effect of N_OD_ on surface roughness.

**Figure 10 materials-12-01940-f010:**
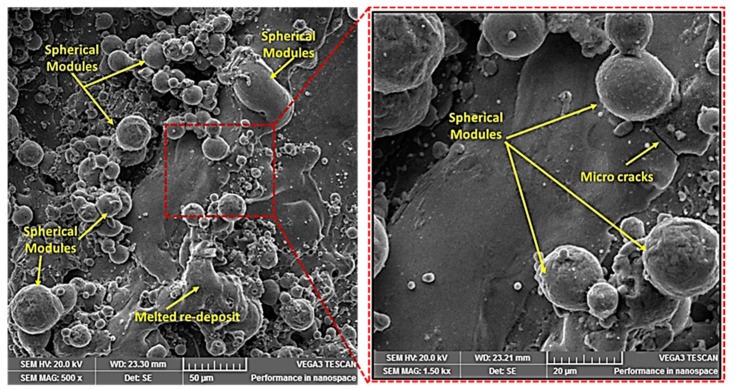
SEM of machined specimen at 200 mm N_OD_.

**Figure 11 materials-12-01940-f011:**
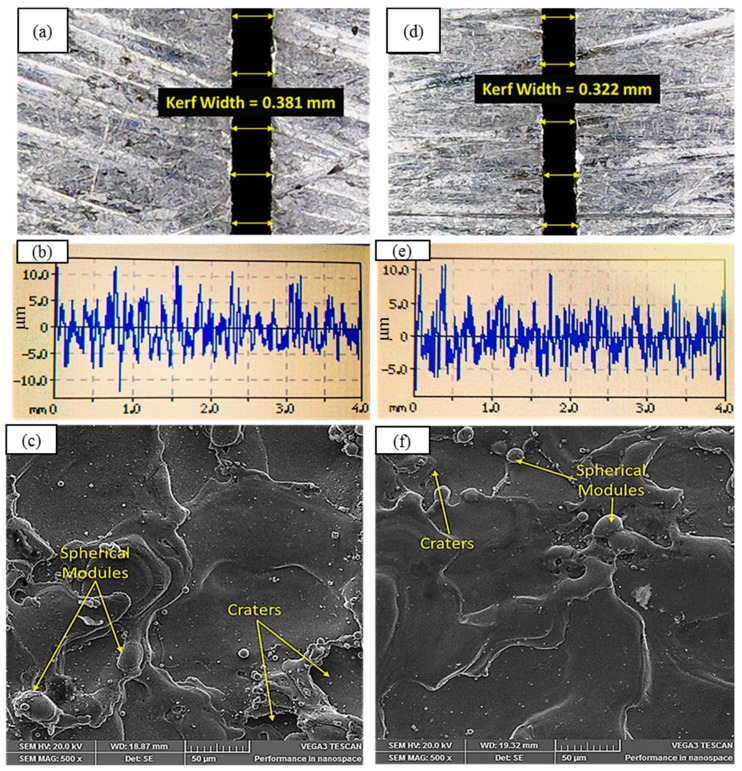
Kerf width, surface roughness, and SEM micrographs of machined surface under: (**a**–**c**) nonoptimal settings—V = 70 V, DS = 35 Hz, I = 3 A, and N_OD_ = 240 mm; and (**d**–**f**) optimal setting—V = 50 V, DS = 35 Hz, I = 3 A, and N_OD_ = 220 mm.

**Table 1 materials-12-01940-t001:** Chemical composition of stainless steel 304.

Elements	C	Cr	Si	Ni	S	P	Fe
Weight (%)	0.075	18.4	1.03	9.76	0.03	0.048	Balance

**Table 2 materials-12-01940-t002:** Input parameters and their levels.

Input Parameters	Units	Levels			
		1	2	3	4
**Voltage (V)**	V	50	60	70	80
**Drum Speed (DS)**	Hz	35	40	45	50
**Current (I)**	amp	1	2	3	4
**Nozzle Offset Distance (N_OD_)**	mm	200	220	240	260

**Table 3 materials-12-01940-t003:** ANOVA for surface roughness (SR), cutting speed (CS), and kerf width (KW).

**ANOVA for SR**							
**Source**	**DF**	**Seq SS**	**Adj SS**	**Adj MS**	**F**	**P**	**% Contribution**
**V**	3	2.22887	2.22887	0.74296	25.69	0.012	45
**DS**	3	1.27454	1.27454	0.42485	14.69	0.024	26
**I**	3	0.32664	0.32664	0.10888	3.77	0.153	7
**N_OD_**	3	1.02303	1.02303	0.34104	11.79	0.036	20
**Error**	3	0.08675	0.08675	0.02892			2
**Total**	15	4.93983					
**S = 0.170050**		**R − Sq = 98.24%**		**R − Sq (adj) = 91.22%**			
**ANOVA for CS**							
**Source**	**DF**	**Seq SS**	**Adj SS**	**Adj MS**	**F**	**P**	**% Contribution**
**V**	**3**	**0.28033**	**0.28033**	**0.09344**	**3.56**	**0.163**	**6**
**DS**	3	0.27686	0.27686	0.09229	3.51	0.165	6
**I**	3	4.10536	4.10536	1.36845	52.08	0.004	85
**N_OD_**	3	0.05902	0.05902	0.01967	0.75	0.591	2
**Error**	3	0.07883	0.07883	0.02628			1
**Total**	15	4.80039					
**S = 0.162098**		**R − Sq = 98.36%**		**R − Sq (adj) = 91.79%**			
**ANOVA for KW**							
**Source**	**DF**	**Seq SS**	**Adj SS**	**Adj MS**	**F**	**P**	**% Contribution**
**I**	3	0.01624	0.01624	0.00541	108.19	0.001	53
**N_OD_**	3	0.00090	0.00090	0.00030	6.06	0.087	3
**V**	3	0.00671	0.00671	0.00224	44.71	0.004	22
**DS**	3	0.00647	0.00647	0.00216	43.14	0.006	21
**Error**	3	0.00015	0.00015	0.00005			1
**Total**	15	0.03047					
**S = 0.007073**		**R − Sq = 99.51%**		**R − Sq (adj) = 86.00%**			

**Table 4 materials-12-01940-t004:** Grey relational generating.

Exp No.	Grey Relational Generating			Grey Relational Coefficient			GRA Grade	Ranking
	SR	KW	CS	SR	KW	CS		
**1**	0.0000	0.7980	0.1965	0.3300	0.7120	0.3835	0.4751	14
**2**	0.2940	0.5710	0.5648	0.4140	0.5380	0.4696	0.4738	15
**3**	0.3880	0.4280	0.6796	0.4490	0.4660	0.6095	0.5081	11
**4**	0.6700	1.0000	0.7536	0.6020	1.0000	0.6699	0.7572	2
**5**	0.6260	0.3700	0.6250	0.5720	0.4270	0.4820	0.4936	12
**6**	1.0000	0.0840	0.0194	1.0000	0.3530	0.3377	0.5635	8
**7**	0.6090	0.4410	0.7702	0.5610	0.4720	0.6851	0.5726	7
**8**	0.5180	0.4610	0.6140	0.5090	0.4810	0.7799	0.5899	6
**9**	0.4170	0.6880	0.8640	0.4610	0.6000	0.7862	0.6157	3
**10**	0.4710	0.3830	0.8113	0.4850	0.4490	0.7260	0.5533	10
**11**	0.4700	0.4740	0.2552	0.4850	0.4870	0.4017	0.4579	16
**12**	0.3820	0.7920	0.7400	0.4470	0.7060	0.6579	0.6036	4
**13**	0.1530	0.0000	0.8477	0.3710	0.3300	0.7665	0.4891	13
**14**	**0.7530**	**0.7660**	**1.0000**	**0.6690**	**0.6810**	**1.0000**	**0.7833**	**1**
**15**	0.7410	0.7460	0.4271	0.6580	0.6630	0.4859	0.6022	5
**16**	0.7150	0.7920	0.0000	0.6360	0.7060	0.3330	0.5583	9

**Table 5 materials-12-01940-t005:** Results of confirmatory test.

Sr. No.	Input Parameter’s Setting	Parameter’s Level	Level Values	Cutting Speed (mm/min)	Kerf Width (mm)	Surface Roughness (µm)
**1**	Optimal Settings	V 1, DS 1, I 3, N_OD_ 2	V = 50 V, DS = 35 Hz, I = 3 A, N_OD_ = 220 mm	2.62	0.322	4.47
**2**	Nonoptimal Settings	V 3, DS 1, I 3, N_OD_ 3	V = 70 V, DS = 35 Hz, I = 3 A, N_OD_ = 240 mm	2.02	0.374	5.69
**Percentage Improvement**				29%	16%	27.3%

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
