# Peer review of "Evaluating Material’s Interaction in Wire Electrical Discharge Machining of Stainless Steel (304) for Simultaneous Optimization of Conflicting Responses"

_materials, 2019, doi:10.3390/ma12121940_

Round 1
Reviewer 1 Report
The manuscript is interesting, written in good English and the figures and tables are relevant and correct.
Still minor corrections needed:
In Abstract ‘ANOVA’ is not explained – it is explained only in page 4.
In Introduction is given a strange unit Cº
In page 2 an abbreviation ‘MMR’ is not defined.
In page is not defined abbreviation ‘CNC’.
In page 3 the steel density is given in the imperial units as in everywhere else the meter units are used; also ‘68HRB’ and ‘0.18mm’ need a space.
There is written that “…composition of SS 304 was obtained by spectroscopy method...” – specify what method.
Surface roughness was measured by “…Surface Texture Meter (S128)”, it would be nice add what company produced the meter (Surtronic?). Also, how many profiles was measured on a sample to get its surface roughness value?
In the beginning of § 3 is marked that in analysis are used ANOVA and GRA methods, but in § 2 the authors must explain in which form they explored these methods – did they used any special program, or used some algorithms/tables in web, or did calculated manually using formulas. If any program or web-calculator or used must be given its name and version.
In page 4 given contribution values are hardly so exact that to give them with hundredths of percentage precision.
Reviewer 2 Report
The authors present a optimization study on the SS304 steel in wire electric discharge machining. There are many parameters and data involved in the manuscript, but the main concern is the results are not clearly presented and should be revised before publication. 1. The full name of "WEDM" is required in the title. 2. Some background and method content in the Results should be mentioned or moved to the Methods section to present the result in a concise way. 3. How many duplicates were used for each tests? Error bars should be included in the Fig. 3, 5, 8, 9. 4. It is not a good presentation using the "optimal settings" and "other settings" in Table 5, Fig. 11-13. 5. Some data should be considered to be combined in a concise way.Author Response
Please see the attached file

Round 2
Reviewer 2 Report
Accept in present form.